# Nanoparticle-Based Drug Delivery for Vascular Applications

**DOI:** 10.3390/bioengineering11121222

**Published:** 2024-12-03

**Authors:** Atanu Naskar, Sreenivasulu Kilari, Gaurav Baranwal, Jamie Kane, Sanjay Misra

**Affiliations:** Vascular and Interventional Radiology Translational Laboratory, Department of Radiology, Mayo Clinic, Rochester, MN 55905, USA; naskar.atanu@mayo.edu (A.N.); kilari.sreenivasulu@mayo.edu (S.K.); baranwal.gaurav@mayo.edu (G.B.); kane.jamie@mayo.edu (J.K.)

**Keywords:** nanoparticle, drug delivery, vascular applications, biomedical applications, endothelium

## Abstract

Nanoparticle (NP)-based drug delivery systems have received widespread attention due to the excellent physicochemical properties of nanomaterials. Different types of NPs such as lipid NPs, poly(lactic-co-glycolic) acid (PLGA) NPs, inorganic NPs (e.g., iron oxide and Au), carbon NPs (graphene and carbon nanodots), 2D nanomaterials, and biomimetic NPs have found favor as drug delivery vehicles. In this review, we discuss the different types of customized NPs for intravascular drug delivery, nanoparticle behaviors (margination, adhesion, and endothelium uptake) in blood vessels, and nanomaterial compatibility for successful drug delivery. Additionally, cell surface protein targets play an important role in targeted drug delivery, and various vascular drug delivery studies using nanoparticles conjugated to these proteins are reviewed. Finally, limitations, challenges, and potential solutions for translational research regarding NP-based vascular drug delivery are discussed.

## 1. Introduction

The rapid emergence of nanotechnology and continuous research on nanomaterials over the last two decades have opened numerous avenues for various biomedical applications such as imaging [1], biosensors [2], and targeted drug delivery [3]. The use of nanomaterials for drug delivery can be attributed to their unique physicochemical properties, including controlled and sustained drug release and bioavailability [4]. Additionally, the size (1–100 nm) of nanoparticles (NPs) enables them to cross cell membranes and deliver therapeutic materials to intracellular target sites. Ligand-conjugated or polymer-coated NPs allow for improved biocompatibility and precision in targeted drug delivery [5,6]. NPs are used for clinical diagnosis as contrast agents in biomedical imaging and to detect specific biomolecules in biological samples [1,2]. There are over 90 medicines based on NPs that have been approved for clinical use, which shows their sheer magnitude in todays’ medical industry [7].

In targeted drug delivery, NPs must cross biological barriers, including the vascular endothelium, and must have the capability to withstand enzymatic degradation in the circulation before it they reach the desired target site [8]. In addition, the vascular endo-thelium is the first barrier for NP- based drug delivery during intravenous administration [9]. The fate of NPs in the blood vessels also depends on, (1) margination to the vascular wall, (2) adhesion to the endothelium, and (3) uptake by the endothelial cells. The anionic surface of the endothelium (due to glycosaminoglycans, glycoproteins, and glycolipids) and the cationic surface charge of the NPs are critical for NP– and endothelium interac-tions, and better efficacy of achieving targeted drug delivery with better efficacy [10].

In this review, we will briefly introduce the types of NPs used as drug delivery agents. Next, we will discuss NP behavior and their fate in blood vessels. Subsequently, the required factors of nanomaterials for vascular applications will be elaborated. Target-ing strategies for nanomedicines will also be briefly discussed, including the effect of in-flammation on these strategies. Penultimately, different nanoplatform- based vascular drug delivery applications will be discussed. Finally, the prospects, challenges, and po-tential solutions regarding NP- based vascular drug delivery will be described.

## 2. Types of Nanoparticles Used as Drug Delivery Agents

The advancement and utilization of nanotechnology for biomedical applications such as drug delivery have grown in the past couple of decades due to the advantageous physiochemical properties of nanomaterials [11]. Researchers have utilized various NPs such as liposomes [12], polylactide (PLA) [13], PLGA [14,15], metal NPs [16], and carbon-based NPs [17] as drug delivery vehicles for vascular drug delivery.

Among the lipid-based NPs are liposomes, solid lipid NPs (SLNs), and nanostructured lipid carriers (NLCs), which are known for their excellent drug-carrying ability [18]. Liposomes were the first nanomedicine to obtain Food and Drug Administration (FDA) approval in the liposomal doxorubicin (DOX) delivery system Doxil^®^ in 1995 [19]. Liposomes possess a morphology like that of cell membranes and mainly consist of phospholipids with both hydrophilic and lipophilic properties, which enables them to carry and deliver hydrophilic, hydrophobic, and lipophilic compounds [20]. Additionally, SLNs are utilized as drug delivery vehicles due to their large surface areas and their ability to carry both hydrophobic and hydrophilic drugs. However, the drug loading capacity of SLNs is low [21]. In contrast, NLCs have a higher drug loading capacity, with better stability and lower drug expulsion during storage. NLCs are therefore a highly promising option for vascular drug delivery [22]. However, their application protein and peptide drug delivery need to be investigated further.

Poly lactic-co-glycolic acid (PLGA) NPs are another type of FDA-approved NP that are widely used for drug delivery due to their biocompatibility and effective biodegradability [23]. PLGA NPs can incorporate hydrophilic and hydrophobic therapeutic agents with the help of emulsion solvent diffusion methods, enabling injectable hydrogel-based vascular drug delivery [14,15]. However, their hydrophobicity and their initial burst of drug release require further investigation.

Dendrimers are polymeric globular branched molecules and are the smallest of all the nanocarriers (sizes vary from 1 to 15 nm) with near-monodisperse complex three-dimensional structures [24]. They have excellent potential as a drug delivery vehicle due to favorable properties such as their nano size, biocompatibility, bioavailability, solubility, permeability, and interactions with membranes. Moreover, their exterior surfaces can be functionalized with ligands or antibodies for targeted drug delivery [25].

Inorganic NPs, such as gold (Au), silver (Ag), iron oxide nanoparticles (IONPs), and mesoporous silica nanoparticles (MSNPs), have attracted tremendous attention for drug delivery and biomedical imaging due to their distinctive physical, electrical, optical, and magnetic properties [26]. Inorganic NPs can be functionalized during their synthesis, which reduces the chances of agglomeration and toxicity and enhances their drug retention ability [27]. The physiochemical properties of inorganic NPs are highly corelated with their sizes, shapes, and surface functionalization [28]. However, the in vivo toxicity, biodegradability, and efficacy of IONPs remain unknown and require further evaluation.

Carbon-based NPs such as graphene and carbon nanotubes (CNTs) have also been utilized in bioimaging, biosensing, and drug delivery due to their favorable physicochemical properties [29]. CNTs generally carry drugs in their inner spaces, and their outer surfaces are functionalized with biological molecules for targeted drug delivery [30]. CNTs can be exploited to address issues like the poor solubility of drugs and sustained drug release. Graphene is a single layer of carbon arranged in a honeycomb lattice and is another carbon-based material that is the subject of many studies related to biomedical applications [31]. The high surface area of graphene, along with ample binding sites for biomolecules to interact with, makes it an excellent drug delivery vehicle.

Recently, 2D nanomaterials like black phosphorus (BP) [32] and molybdenum disulfide (MoS_2_) [33] have generated enormous attention after the huge success of graphene in targeted drug delivery due to their biocompatibility and biodegradability. The excellent photoactive (photothermal and photodynamic) nature of BP is due to its broad absorption across the spectrum of UV, visible, and infrared light. Along with the visible-light (400–750 nm) absorption of MoS_2_, its high surface area, biocompatibility, and biodegradability make BP promising for potential drug delivery applications [34].

Cell membrane-coated NPs (CMNPs) are another type of innovative nanomaterial that basically cloaks a synthetic nanoparticle with a natural cell membrane, which enables it to mimic the biological characteristics of the cell membrane while keeping the physicochemical properties of the nanomaterial itself [35]. CMNPs have several advantages over liposomes, such as selective permeability and biocompatibility [36]. Some of the major advantages of CMNPs are high target specificity, long-term blood circulation, and excellent biocompatibility. Cell membranes from red blood cells (RBCs) [37], white blood cells (WBCs) [38], platelets [39], extracellular vehicles (EVs) [40], stem cells [41], and the outer membrane vesicles of bacteria [42] have been utilized for NP encapsulation. The primary objective of these types of nanoplatforms is to evade immune clearance and maintain a long circulation time.

Biocompatibility and biodegradability, along with drug release potential, are the primary factors in selecting the types of nanoparticles for drug delivery applications [43]. FDA-approved PLGA NPs are mostly utilized for drug delivery applications due to their physicochemical properties, along with their biocompatibility and biodegradability [23,44]. Besides PLGA NPs, liposomes are other nanomaterials that have been widely used for drug delivery applications [20]. Liposomes are considered highly biocompatible compared to other synthetic materials because the biophysical and biochemical properties of liposomes are similar to those of cell membranes [43]. Inorganic NPs such as Au, Ag, IONPs, and MSNPs have great potential for drug delivery applications and have been the subjects of several studies. The toxicity of these nanomaterials remains debatable and may prevent their usage in clinical applications. Inorganic NPs can be functionalized for better biocompatibility compared to other nanomaterials [27]. Toxicity and biodegradability remain as drawbacks for carbon-based nanomaterials [18].

## 3. Nanoparticle Behaviors in Blood Vessels

The fate of NPs during in vivo transport depends on interactions between the NPs and vascular components such as RBCs, WBCs, platelets, plasma proteins, and bioactive factors of the vascular endothelium after they enter the blood vessels [45]. Moreover, the NPs must follow sequential processes of endothelial uptake such as margination and adhesion. Therefore, it is important to understand NP behavior and its effect on endothelial uptake and targeted applications, including their stability in the blood stream [46].

### 3.1. Blood Flow

The rheological characteristics of blood flow substantially influence the fate of NPs after entering the blood circulation. Due to the heavy blood flow and varying shear rates induced by flow velocities, NPs barely manage to diffuse, which hinders their migration and distribution in the vascular endothelium [9]. In addition to the blood flow, NP interactions also depend on NP properties (such as size, shape, and surface characteristics) and the characteristics of the vasculature [47]. For example, Ye et al. [48] computationally showed that large particles or geometric forms have the capability to overcome being trapped by an RBC core compared to spherical shapes.

### 3.2. Margination

Margination is defined as the ability of suspended NPs or cells to escape the blood flow and move toward the vascular endothelium wall [49]. In a blood vessel, RBCs generally migrate away from the vascular wall due to the sheer force of the blood flow and create a cell-free layer. Meanwhile, NP size is also a determinant factor regarding their marginalization, as vigorous Brownian motion is associated with smaller NPs and determines their fate [49]. Regarding this, smaller particles like NPs prefer to migrate toward the cell-free layer of the vascular wall [50]. NPs with higher margination abilities frequently interact with the vascular endothelium, resulting in more effective adhesion and uptake. Moreover, the sizes, shapes, and surface characteristics of NPs also determine their margination abilities [50]. For example, Kona et al. [51] showed that endothelial cells (ECs) preferred to take up smaller particles (200 nm–1 μm) compared to larger sizes.

### 3.3. Adhesion

Adhesion is the process by which NPs relocate onto the endothelial layer after their margination from the center of the blood vessel [52]. The margination and adhesion steps are closely related to each other, but the adhesion step requires some external forces such as van der Waals interactions, electrostatic interactions, and some surface modifications of the NPs for the interactions between the NPs and the vascular surface [53]. However, blood flow-induced shear stress can also remove adhered NPs [54]. Taken together, the surface charges of the NPs and the endothelial layer play a vital role in mediating the adhesion of the NPs to the endothelium.

### 3.4. Uptake by Endothelium

Uptake by the endothelium is the final step in NP movement out of the blood vessel. This process is also affected by various factors such as blood flow-induced shear stress, NP properties, and the morphological and functional properties of different ECs [55,56]. For instance, Chen et al. [57] showed that NP uptake was more efficient with a reduced flow rate compared to an increased flow rate using an intelligent microfluidic system with a controllable flow rate. Following the same principle, Zhang et al. [58] revealed that NPs prefer to distribute in low shear stress regions compared to those with high shear stress. Additionally, the inflammatory state of the vascular wall and its related expression of adhesion molecules (e.g., VCAM-1) potentially improve the uptake of vascular-targeted drug delivery [59].

## 4. Required Factors of Nanomaterials for Vascular Applications

### 4.1. Particle Size and Shape

Particle size is one of the most extensively studied properties of nanoparticles due to its immensely important role in drug delivery [60]. Due to their high surface area to volume ratios, nanomaterials have large surface areas for interaction. Additionally, the sizes of nanomaterials allow them to easily penetrate body tissues and fluids. It is also clear from various studies that the sizes of nanoparticles determine their fates in cell systems due to factors like the rate of cellular uptake and the uptake mechanism [4]. For example, the size of the nanoparticles in a drug delivery system is usually in the range of 20–200 nm to improve blood circulation, vessel penetration, and biodistribution. It is noteworthy that the kidneys promptly filter NPs smaller than 20 nm, while the reticuloendothelial system easily captures and clears particles larger than 200 nm [9]. For immune cells such as macrophages, nanoparticles smaller than 500 nm enter through the phagocytotic pathway [61].

Additionally, external force is required to help NPs to the edges of blood vessels, as smaller NPs have more active Brownian motion, which hinders their ability to follow the direct marginalization route. For instance, Li et al. [62] showed that comparatively small NPs (≤100 nm) are often transported with RBCs in the blood flow, which reduces their chances of accumulation at the vascular site, while large particles (>100 nm) accumulate near the vascular endothelium wall due to their better marginalization ability [63]. Furthermore, NP adhesion onto the vascular endothelium is also driven by their size. Particles smaller than 200 nm show more prominent characteristics of adhesion to the endothelium after margination compared to larger particles [64]. In a similar manner, particle size also determines endothelial uptake. Particles smaller than 200 nm have a better chance of endothelial uptake than those with larger sizes [51]. Particle size also plays an important role in intracellular localization. In this respect, Oh et al. [65] showed that 2.4 nm Au NPs were localized in the nucleus, whereas larger particles (up to 89 nm) were localized in the cytoplasm after internalization. Hence, it is important to synthesize nanoparticles with tunable size properties.

In addition to the tunable sizes of nanoparticles, shape is another crucial factor that determines their fates as drug delivery vehicles. Particle shape can influence margination behavior in the blood vessel [49]. For example, Decuzzi et al. [66] showed that particles with different shapes (rod/hemisphere/disc) displayed higher margination abilities at the vascular wall than spherical ones. A change in shape from a spherical to a non-spherical structure resulted in a larger contact area for particles when adhering to the endothelium and reduced the drag resistance force, which gave the particles more stability on the vascular wall surface [67]. Similarly, Namdee et al. [68] exhibited a 31% increase in margination to the vessel wall with 1–2 μm rod-shaped particles compared to spherical particles.

### 4.2. Surface Functionalization

The surface charge is another important factor for nanoparticles that can decide their pathway in a cell system [9]. There is more interaction between the endothelium and cationic nanoparticles compared to negatively or neutrally charged particles [69]. This may explain the limited permeability of rod-shaped Au NPs into cells with a negative surface charge [70,71]. Therefore, recent studies have utilized cationic liposomes for electrostatic conjugation with NPs to facilitate their transport into cells.

Surface coating with a polymer or the use of some other nanomaterials can also alter the properties of nanocomposites used for biomedical applications. For example, poly methyl acrylic acid (PMAA) was surface-coated with ZnO, which reduced its potential cytotoxicity but retained its UV-protection characteristics [72]. Hence, it may be used in cosmetic products. In another example, a Ag NP-functionalized titanium implant surface was able to prevent postoperative infection due to resistant strains of *Staphylococcus epidermidis* and *Staphylococcus aureus* [73]. Monoclonal antibodies can also be conjugated with nanoparticles for more flexible delivery targeting. This can be attributed to the countless number of unique receptors or surface antigens against which antibodies can be developed. Similarly, the conjugation of nanoparticles with endothelial target ligands or modification with endothelial target ligands, such as vascular cell adhesion molecule-1 (VCAM-1), intercellular adhesion molecule 1 (ICAM-1), and platelet endothelial cell adhesion molecule 1 (PECAM-1), also facilitates the targeting of nanoparticles to the vascular endothelium [74].

Additionally, biocompatibility and release kinetics are some other factors, along with the surface functionalization, size, and shape of the NPs, that must be considered for effective drug delivery applications.

## 5. Targeting Strategies for Nanomedicines

Targeting strategies for NP-based drug delivery are generally categorized into passive targeting and active targeting [75]. The physicochemical properties of a cell broadly determine the passive targeting of NP-based drug delivery. For example, the local vascular permeability and inflammation responses, which are normally increased by the release of cytokines from damaged tissues, enhance the permeability and retention (EPR) effect [76]. In contrast to passive targeting, the active targeting strategy uses an approach where NPs are conjugated to the ligands of surface determinants specific to the area of interest [77]. Immunostaining, PCR, Western blotting, and flow cytometry are some of the useful techniques used to detect proteins that are enriched in the tissues of interest. Additionally, after numerous efforts by researchers in the field of vascular delivery, certain endothelial surface molecules have emerged as targets for drug delivery. Ideal target determinants can be defined as determinants that would be selectively expressed and accessible at the target site, which include enzymes, cell adhesion molecules (CAMs), integrins, receptors, and transporting molecules.

Endothelial cells undergo many changes under pathological conditions. For example, the intercellular adhesion molecule 1 (ICAM-1)/CD54 level was enhanced in the endothelia during inflammation in murine models of endotoxemia [78]. Similarly, Aminopeptidase N (APN)/CD13, Tumor Endothelial Marker 1 (TEM-1)/CD248, Vascular Cell Adhesion Molecule 1 (VCAM-1)/CD106, E-selectin/CD62E, and P-selectin/CD62P are some of the common inducible markers expressed by pathological endothelia [74]. Furthermore, these determinants are specific to pathologically altered endothelia, which makes them preferable determinants for diagnostic imaging and therapeutic interventions.

## 6. Nanoparticle-Based Vascular Applications

In recent years, NPs were increasingly exploited for targeted drug delivery especially vascular applications. Different types of NPs and their excellent vascular applications are listed in Table 1.

### 6.1. Lipid NPs

Liposomes were the first nanomedicine to obtain FDA approval due to their many advantages, including biocompatibility, high bioavailability, and the ability to carry large volumes of drugs, which are coupled with favorable physicochemical properties for drug delivery [20]. These benefits demonstrate the potential for lipid NPs in vascular drug delivery applications. For example, Chan et al. [79] demonstrated paclitaxel encapsulation with lipid–polymeric nanoparticles for delivery to injured vasculature. In arterial stenosis, compared to sham-injury groups, this treatment showed a ∼50% reduction in arterial restenosis. In another work, Zhao et al. [108] showed the importance of metformin, an anti-diabetic drug, in its polymeric form (PolyMet) in combination with lipid NPs. This formulation systemically delivered vascular endothelial growth factor (VEGF) siRNA for VEGF knockdown in a human lung cancer xenograft. Additionally, degradable lipid nanoparticles were also utilized for potent gene knockdown in multiple biological targets (including hepatocytes and immune cells) [109]. Akhlaghi et al. [80] showed a green formulation approach for preparing curcumin-loaded lipid-based nanoparticles for the inhibition of postangioplasty restenosis. The expression of Ki67 was markedly lower in the curcumin-based lipid nanoformulation group, which verified its potential to prevent neointimal hyperplasia. Liposomal NPs were utilized to deliver clodronate, which inhibited neointimal growth in a balloon-injured rabbit carotid artery [81].

### 6.2. PLGA NPs

PLGA NPs are another type of FDA-approved NP, and a plethora of biomedical applications have been studied [23]. Owing to excellent physicochemical properties such as biocompatibility and biodegradability, these NPs are commonly used for the inhibition of restenosis [15]. Previously, drug-eluting stents (DESs) were the primary mode of treatment during stenosis. However, their low cell-type specificity and their delayed responses to heal the endothelium often led to inflammatory complications [110]. In recent times, drug-eluting balloons (DEBs) have emerged as an alternative to DESs [111]. In this scenario, nanomaterials, specifically PLGA NPs, have shown excellent potential as transport vehicles for targeted delivery. A PLGA NP-based drug delivery platform showed the potential to inhibit restenosis and decrease late-stage adverse events following vascular angioplasty [14,15].

For example, our group showed the importance of encapsulation of 1α,25(OH)_2_D_3_ in PLGA nanoparticles, which inhibited venous neointimal hyperplasia and stenosis in porcine arteriovenous fistulas [14]. Wang et al. [112] developed a multilayer stent with PLGA as a drug-eluting layer that releases sirolimus. Additionally, the release of the drug from PLGA occurs in two stages: diffusion-controlled delivery and degradation-controlled delivery. In another report, PLGA/amorphous calcium phosphate (PLGA/ACP) was used for stent coating, and a combination of paclitaxel and sirolimus could be almost completely eluted within 21 days in vitro and in vivo [82]. Nakano et al. [83] explored the use of cationic PLGA NPs in chitosan-mediated DESs for excellent drug delivery applications. Similarly, PLGA NPs were used to encapsulate 1α, 25-dihydroxyvitamin D3 for adventitial delivery, which attenuated restenosis in a murine angioplasty model [113].

Gholizadeh et al. [84] synthesized PLGA-PEG nanoparticles for targeted delivery of the mTOR/PI3 kinase inhibitor dactolisib in TNF-α-activated endothelial cells, which makes it an interesting nanomedicine for anti-inflammatory therapies. In a similar experiment, Galindo et al. [85] were able to make peptide-targeted PLGA-PEGylated NPs for loading Licochalcone-A, which is a drug used to treat ocular inflammation. Doxorubicin-loaded PLGA NPs were used to target ICAM-1 in lung epithelial cells [86]. In another study, negatively charged PLGA NPs were coupled with highly positively charged polyethylenimine (PEI) to deliver an angiogenesis-related peptide (apelin) and a gene (vascular endothelial growth factor (VEGF)_165_) for neoangiogenesis of human mesenchymal stem cells [87].

Cell membrane-coated PLGA NPs have gained widespread interest for vascular-based drug delivery and other biomedical applications. For example, macrophage membrane (MM)-coated rapamycin-loaded PLGA NPs (RAPNPs) (Figure 1) demonstrated good biocompatibility and efficiently inhibited phagocytosis by macrophages, which targeted activated endothelial cells in vitro through interactions between the integrin α4β1 surface protein on MMs and overexpressed VCAM-1 in ECs. Furthermore, the NPs successfully targeted and accumulated in atherosclerotic lesions in vivo in anti-atherosclerosis applications [88]. In a similar way, Park et al. [114] explored genetically engineered cell membranes coated with dexamethasone-encapsulated PLGA nanoparticles for targeted delivery to inflamed lungs. Golub et al. [115] also showed the potential of PLGA NPs for sustained VEGF delivery to promote vascular growth. Similarly, Lee et al. [116] utilized a layer-by-layer approach to design PLGA NPs encapsulated with heparin and glutathione due to their anticoagulant and antioxidant properties. Hyaluronic acid modification enabled targeting of human bone marrow-derived mesenchymal stem cells (hBMSCs) due to hBMSC homing in I/R injury sites and thus enhanced the overall vascular therapeutic effect.

PLGA NPs are among the most utilized NPs for inhibiting restenosis. In this regard, Palumbo et al. [89] explored the encapsulation of dexamethasone dipropionate into α-Elastin-g-PLGA NPs for the treatment of restenosis. In a new experimental setup, Yang et al. [90] developed a stent coated with bi-layered PLGA NPs in which a VEGF plasmid is in the outer layer and paclitaxel (PTX) is in the inner core. The concept of this experimental setup was to induce re-endothelialization by early release of the VEGF gene alongside slow release of PTX to suppress smooth muscle cell proliferation, which ultimately prevented restenosis. In a similar study conducted by Zhu et al. [91], PLGA NPs loaded with PTX and NaHCO_3_, as a pH-sensitive therapeutic agent, showed the potential to inhibit vascular restenosis. Rapamycin [117] and lisinopril [118] were also utilized with PLGA NPs for anti-restenosis applications.

### 6.3. Dendrimer

Dendrimers can conjugate bioactive molecules with some ligands or antibodies on their exterior surfaces due to the presence of functional groups on their peripheries, which allows targeted drug delivery [24]. They can also be loaded with drugs with poor solubility. For dendrimer-based vascular drug delivery applications, Fu et al. [119] showed the potential of dendrimer-based NPs with PEG for iodinated contrast agent delivery, which is compatible with CT imaging. These dendrimer-based contrast agents showed potential applications in quantitative microvascular characterization of disease states like ischemic injury, inflammation, and cancer. Moreover, Xu et al. [92] utilized a folic acid-decorated polyamidoamine dendrimer as a vector for local delivery of siRNA against vascular endothelial growth factor A (siVEGFA) in a xenograft tumor mouse model, resulting in a profound reduction in angiogenesis. An arginine-glycine-aspartic peptide (RGD)-modified polyamidoamine (PAMAM G1) dendrimer showed excellent potential for the prevention of restenosis [93]. It is noteworthy that dendrimers are mostly used for cell labeling, rather than treatment [120].

### 6.4. Inorganic NPs

Among the inorganic NPs, Au NPs, IONPs, and MSNPs are some of the most widely used NPs for drug delivery. Au NPs were synthesized in varied sizes and shapes, such as spheres, prisms, cubes, rods, and cages. They also showed excellent physicochemical properties such as biocompatibility, tunable optical properties, and easy functionalization for drug delivery applications [121]. In terms of vascular applications, Khoobchandani et al. [94] explored the therapeutic potential of epigallocatechin-3-gallate-conjugated gold nanoparticles (EGCg-AuNPs) as a possible alternative to drug-coated stents for the proliferation and migration of human smooth muscles cells (SMCs) and endothelial cells (ECs). The nanocomposite also showed a favorable toxicity profile for vascular drug delivery applications. In another approach, fibronectin (FN) was incorporated into gold nanoparticles (AuNPs) and coated onto catheters with mesenchymal stem cells (MSCs) for vascular tissue regeneration [95]. Similarly, Meyers at al. [122] used a surface-functionalized Au NP with a collagen-targeting peptide to bind to sites of arterial injury following vascular interventions. This functionalized and biocompatible nanoparticle targets vascular injuries following systemic administration. In another instance, Sun et al. [96] showed the importance of VCAM1-binding Au nanospheres for targeted delivery of anti-miR-712 into the inflamed endothelium for the purpose for atherosclerosis therapy.

IONPs are widely used inorganic NPs for drug delivery applications due to their excellent physicochemical properties. In respect to the vascular drug delivery application, monoclonal anti-VCAM-1 antibody-conjugated multifunctional core–shell Fe_3_O_4_@SiO_2_ nanoparticles were synthesized for selective adhesion and delivery to endothelial cells [97]. A monoclonal anti-VCAM-1 antibody was utilized as a targeting ligand due to the overexpression of VCAM-1 receptors in inflamed endothelial cells. Similarly, intercellular adhesion molecule-1 (ICAM-1) antibody-conjugated IONPs were synthesized to target the overexpression of ICAM-1 in human triple-negative breast cancer cell lines and were utilized as magnetic resonance imaging (MRI) probes [98]. Likewise, a cationic derivative of chitosan (CCh) was utilized to coat IONPs and was conjugated with monoclonal antibodies (anti-VCAM-1 and anti-P-selectin) for magnetic resonance imaging of endothelial inflammation [99]. Anti-CD34-grafted IONPs were also synthesized to demonstrate the feasibility of endothelial progenitor cell (EPC) adhesion on an iron stent for rapid endothelialization [100]. Riegler et al. [123] also showed the potential of IONPs targeting mesenchymal stem cells. This resulted in a six-fold increase in cell retention following balloon angioplasty in a rabbit model and ultimately led to a reduction in restenosis three weeks after cell delivery. Li et al. [101] successfully explored targeted mitigation of neointimal hyperplasia through indocyanine green (ICG)-conjugated SPIONs labeled with EPCs following vascular injury. In addition, these nanoparticles were nontoxic at the chosen concentration. Anti-collagen IV peptide-conjugated IONPs were utilized to deliver rapamycin for vascular restenosis therapy [124]. Dexamethasone phosphate was also conjugated with IONPs due to its ability to accumulate drugs in specified regions of the vasculature in a rabbit model of atherosclerosis [125].

MSNPs are generally nontoxic at sizes <100 nm and concentrations up to 100 μg/mL. MSNPs can also be surface functionalized for effective drug delivery and have therefore been utilized for systemic, localized, and oral delivery. MSNPs have high levels of biocompatibility in a biological system [126]. In this regard, Wu et al. [102] exploited PEGylated Cu-doped MSNPs to encapsulate an anti-inflammatory interleukin-1 receptor antagonist (IL-1Ra) for codelivery of Cu ions and IL-1Ra, which significantly reduced arterial stenosis, the plaque burden, and macrophage infiltration in mice with carotid plaques. Pham et al. [103] showed the potential of CD9 antibody-functionalized, hyaluronic acid-coated MSNPs for targeted delivery of the anti-senescence drug rosuvastatin. This MSNP-based nanoplatform successfully delivered rosuvastatin to senescent atherosclerotic plaques and mitigated atherosclerosis. In another work, a titanium coating on MSNPs improved their biocompatibility and improved the release dynamics of sodium nitroprusside [127]. Vascular restenosis was also inhibited through honokiol-encapsulated MSNPs via suppression of the TGF-β signaling pathway after common carotid artery injury in rats [128].

### 6.5. Carbon Based NPs

Carbon-based NPs have also been widely explored in various biomedical applications like bioimaging, drug delivery, and biosensing. In respect of vascular applications, Paul et al. [129] developed a CNT-coated stent device capable of preventing postangioplasty in-stent restenosis through substantial vascular endothelial recovery in a site-specific manner. In another work, a paclitaxel-eluting porous carbon–carbon nanoparticle-coated nonpolymeric cobalt–chromium stent showed adequate efficiency in endothelialization, reducing neointimal hyperplasia, reducing the percentage diameter of stenosis, and reducing the inflammatory response [130]. Moreover, Ge et al. [104] effectively demonstrated inhibition of in-stent restenosis with a double-layer-coated vascular stent, which included an inner layer of docetaxel (DTX)-loaded graphene oxide (GO) and an outer layer of carboxymethyl chitosan (CMC) loaded with heparin (Hep). A polyethylenimine (PEI)-modified single-walled CNT conjugated with candesartan was developed to deliver VEGF-targeted siRNA (siVEGF) for the synergistic treatment of tumor angiogenesis [105]. Jing et al. [131] also showed the potential for small-diameter vascular graft applications by electrospinning thermoplastic polyurethane/graphene oxide scaffolds.

### 6.6. Two-Dimensional Nanomaterials

The widespread use of graphene has paved the way for other 2D nanomaterials like BP and MoS_2_ for various biomedical applications [32]. Hence, researchers have used these 2D nanomaterials for vascular applications. In this regard, Chen at al. [106] explored the morphological significance of BP quantum dots (BPQDs) and nanosheets (BPNSs) for treating endothelial dysfunction and preventing transcriptome aberrations in mouse aortas. As expected, BPNSs, with irregular shapes and larger lateral sizes compared to BPQDs, were likely to prevent in vitro angiogenesis at non-cytotoxic concentrations and trigger platelet adhesion to HUVECs. Similarly, a MoS_2_-based nanoplatform was utilized for vascular drug delivery applications.

### 6.7. Cell-Membrane Coated NPs (CMNPs)

CMNPs are a recent and innovative approach to NP preparation for various biomedical applications like drug delivery [35], antibacterial activity [132], and bioimaging [133]. Recently, Yao et al. [134] prepared a stem cell membrane-camouflaged nanocomplex via self-assembly of a mesenchymal stem cell membrane on an miRNA-loaded MSNP surface. The advantage of this approach is that it can target ischemic injured cardiomyocytes after escaping clearance by the immunologic system due to its biological membrane. In another example, Li et al. [107] showed the importance of a platelet membrane-coated nanoparticle formation for vascular restenosis reduction (See Figure 2). Interleukin 10 (IL10) nanoparticles can effectively regulate local inflammation, and they were conjugated onto platelet membrane-coated PLGA NPs as a targeted drug delivery system to prevent vascular restenosis.

## 7. Future Perspectives and Conclusions

In this review, we discussed various nanoparticles and their applications as drug delivery agents, including their biological journey in blood vessels. Next, we discussed the factors required for nanomaterials to be effective vascular drug delivery vehicles. After that, various nanoparticle-based vascular applications were thoroughly discussed. The potential of nanoparticles as drug delivery vehicles for vascular drug delivery applications has become clear in recent years. NPs also have the ability to facilitate prolonged release of multiple agents for instant treatment. It is also notable that the number of nanomedicines approved by the FDA is continuously increasing, and the number of ongoing clinical trials based on nanoparticle-based biomedical applications will further add to their potential therapeutic value [61,135,136,137].

Several NP-based drug delivery systems have been utilized to deliver chosen therapeutic agents. However, some of the NP-based drug delivery systems rely too heavily on the inherent properties of the drugs, such as solubility or drug release. Hence, more study is needed to eliminate this dependency on the inherent properties of drugs. This may include deeper utilization of surface functionalization of NPs to evade this issue. Another aspect of NP-based vascular drug delivery that requires proper consideration is rapid blood flow, along with repeated interactions between NPs and numerous blood cells and immune cells. Achieving therapeutic and diagnostic efficacy in this scenario is challenging. The biomimetic principle of cell membrane-coated NPs (CMNPs) has the potential to resolve this issue, as they can evade immune system attacks by minimizing immune recognition to extend circulation time in vivo. They also have a strong targeting ability due to the presence of ligands or proteins on their cell membranes.

Certainly, more attention should be paid to the structural design and surface functionalization of NPs that can target specific ligands that selectively combine with specific molecules. These NPs have excellent drug-loading and retention capabilities and can closely control drug release. Nevertheless, the clinical translation of NP-based vascular drug delivery systems requires more attention due to challenges such as the complex formations of nanocomposites and surface functionalization to reduce toxicity. Moreover, interactions between NPs and pathological tissues need to be properly interpreted. Furthermore, reproducible synthesis of NPs is an essential benchmark for their successful translation to clinical settings.

## Figures and Tables

**Figure 1 bioengineering-11-01222-f001:**
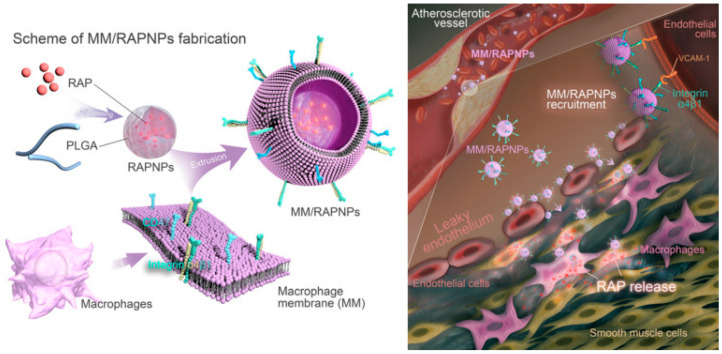
Schematic representation of MM/RAPNP nanoplatform for treatment of atherosclerosis. Reproduced with permission from Ref. [88]. Copyright 2021, Ivyspring International Publisher.

**Figure 2 bioengineering-11-01222-f002:**
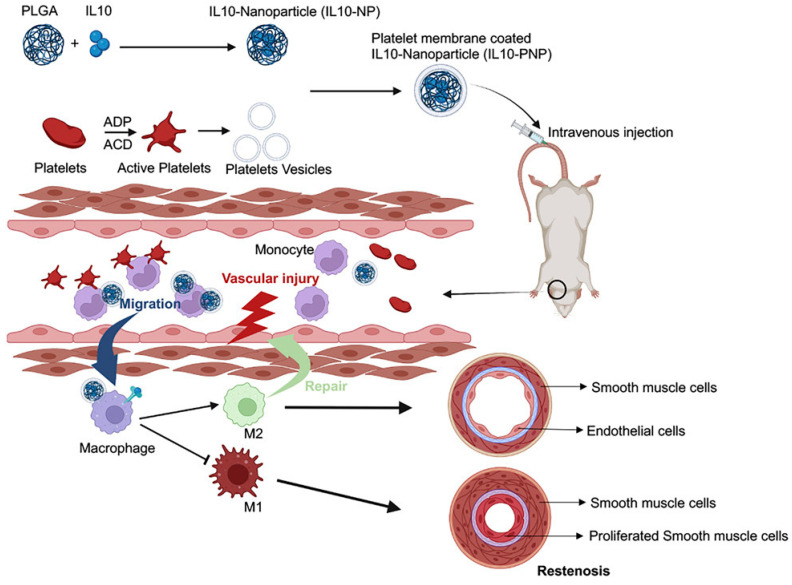
Schematic representation of preparation of platelet membrane coated IL-10 NP for reduction of vascular restenosis. Reproduced with permission from Ref. [107]. Copyright 2022, Elsevier.

**Table 1 bioengineering-11-01222-t001:** Nanoparticle based vascular applications.

Material	Effect	Ref.
Lipid-polymer NPs with PTX	∼50% reduction in arterial stenosis with targeted NP treatment	[79]
CUR NPs	Inhibits of post-angioplasty restenosis	[80]
Liposomal clodronate	Clodronate delivery inactivates and kills macrophages which reduce neointimal hyperplasia and restenosis	[81]
1.25 NPs	Inhibits venous neointimal hyperplasia and stenosis	[14]
PLGA/ACP	PLGA/ACP coating on DES successfully released combined paclitaxel/sirolimus drug for the treatment of coronary arterial diseases	[82]
FITC-PLGA NPs	Bioabsorbable polymeric NP-eluting stent would be an efficient innovative platform for in vivo drug delivery.	[83]
PLGA-PEG NPs	Targeted delivery of dactolisib, the mTOR/PI3kinase inhibitor to inflamed endothelium	[84]
Tet-1 or B6 PLGA-PEG NPs	Targeted delivery of Licochalcone-A drug regarding ocular inflammation.	[85]
cLABL-DOX-PLGA NPs	Successfully targeted ICAM-1-expressing cells and controlled release of drug.	[86]
PEI-PLGA NPs	Delivered an angiogenesis-related peptide (apelin) and a gene (vascular endothelial growth factor (VEGF)_165_) for neoangiogenesis of human mesenchymal stem cells	[87]
MM/RAP PLGA NPs	Biomimetic nanoparticle was used for in vivo atherosclerosis therapy	[88]
α-Elastin-g-PLGA	Dexamethasone dipropionate showed potential for treatment of restenosis	[89]
VEGF/PTX PLGA NPs	Stent coated with bi-layered PLGA nanoparticles promoted re-endothelialization	[90]
PTX-NaHCO_3_-PLGA NPs	pH-responsive nanoplatform showed potential to effectively inhibit restenosis.	[91]
G4-FA	local delivery of siRNA against vascular endothelial growth factor A (siVEGFA)	[92]
RGD-PAMAM G1	Prevents restenosis.	[93]
EGCg-Au NPs	proliferation and migration of human smooth muscle cells (SMCs) and endothelial cells (ECs)	[94]
FN-Au NPs	promoted mesenchymal stem cell proliferation and increased biocompatibility	[95]
VCAM-1- Au Nanospheres	Targeted delivery of anti-miR-712 for atherosclerosis therapy	[96]
VCAM-1-targeted Fe_3_O_4_@SiO_2_(FITC) NPs	Selective adhesion and delivery to endothelial cells	[97]
ICAM-IONPs	Utilized as a magnetic resonance imaging (MRI) probe	[98]
γ-Fe_2_O_3_-CCh-anti-VCAM-1	for endothelial inflammation	[99]
anti-CD34-IONPs	Demonstrated the feasibility of EPC adhesion on an iron stent for rapid endothelialization	[100]
IONPs@PEG-ICG-EPCs	Targeted mitigation of neointimal hyperplasia	[101]
PEGylated Cu-doped MSNs	Significantly reduced arterial stenosis, plaque burden, and macrophage infiltration due to codelivery of Cu ions and IL-1Ra	[102]
CD9-HA-MSNPs	targeted delivery of anti-senescence drug rosuvastatin	[103]
GO/DTX and CMC/Hep	reduced both proliferation and thrombosis	[104]
PEI-modified single-walled CNT–candesartan	VEGF-targeted siRNA delivery for the synergistic treatment of tumor angiogenesis	[105]
BPQDs and BPNSs	Morphological attribute of BP can be crucial for its effect in endothelium	[106]
PM-PLGA-IL10	Targeted delivery of IL-10 to inhibit vascular restenosis.	[107]

Abbreviations: CUR: curcumin loaded lipid-based nanoparticles, PTX: paclitaxel, 1.25 NP: 1α,25(OH)_2_D_3_ in PLGA nanoparticles, ACP: amorphous calcium phosphate, cLABL: cyclo-(1,12)-PenITDGEATDSGC, MM: macrophage membrane, RAP: rapamycin, VEGF: vascular endothelial growth factor, FA: folic acid, G4: polyamidoamine dendrimer G4, PAMAM G1: polyamidoamine, EGCg: epigallocatechin-3-gallate, FN: fibronectin, FITC: fluorescein isothiocyanate, CCh: cationic derivative of chitosan, MRI: magnetic resonance imaging, EPC: endothelial progenitor cell, ICG: indocyanine green, MSNs: mesoporous silica nanoparticles, IL-1Ra: interleukin-1 receptor antagonist, HA: hyaluronic acid, GO: graphene oxide, DTX: docetaxel, CMC: carboxymethyl chitosan, Hep: heparin, PM: platelet membrane, IL-10: interleukin 10.

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
