# Peer review of "Nanoparticle-Based Drug Delivery for Vascular Applications"

_bioengineering, 2024, doi:10.3390/bioengineering11121222_

Round 1
Reviewer 1 Report
Comments and Suggestions for Authors
Dear Editor:
In this review, the authors discussed the different types of customized nanoparticle (NPs) for drug delivery, behaviors of nanoparticle (margination, adhesion, and endothelium uptake) in blood vessels, and nanomaterials compatibility for successful drug delivery in vascular medicine. The cell surface proteins which play an important role in targeted drug delivery and various nanoparticles based vascular drug delivery studies have been reviewed.
In a word, the author need to make minor changes to this review. There are a few suggestions as follow:
1. In module 3, the authors should cite more references to state their points.
2. The module 5 is not relevant and logical enough to the topic.
3. The reference format is not uniform.
4. The fifth and seventh sections have the problem of missing punctuation.
Author Response
â– General statement
In this review, the authors discussed the different types of customized nanoparticle (NPs) for drug delivery, behaviors of nanoparticle (margination, adhesion, and endothelium uptake) in blood vessels, and nanomaterials compatibility for successful drug delivery in vascular medicine. The cell surface proteins which play an important role in targeted drug delivery and various nanoparticles based vascular drug delivery studies have been reviewed.
In a word, the author need to make minor changes to this review. There are a few suggestions as follow:
Response: We appreciate the Reviewer for the recommendation and publish our manuscript for publication in “Bioengineering” after minor revisions.
Comment 1: In module 3, the authors should cite more references to state their points.
Response 1: Thank you very much for this comment. Accordingly, we have added more references in the revised manuscript for the potential readers to understand the topic.
Comment 2: The module 5 is not relevant and logical enough to the topic.
Response 2: Thank you very much to the reviewer for this critical evaluation. As our topic of the review paper is based on nanoparticle-based drug delivery for vascular application, hence we want to discuss the targeting strategies of nanoparticle-based drug delivery. In this regard, active and passive targeting strategies has been discussed along with different determinants. So, the topic in module 5 is discussed in brief which we feel is relevant to the manuscript.
Comment 3: The reference format is not uniform.
Response 3: The references used in the manuscript has been incorporated with the help of latest version of EndNote. Hopefully the uniformity of the references will be published ready in the revised version of the manuscript.
Comment 4: The fifth and seventh sections have the problem of missing punctuation.
Response 4: Thank you very much for this evaluation. Accordingly, we have modified the manuscript and incorporated the missing punctuation. Please see the revised manuscript for this purpose.
Reviewer 2
General statement
Writing reviews is difficult because, operating on data that already exists, it is necessary to show something new. I don't see it in the authors' work. Even such basic things as the division of nanoparticles is chaotic and controversial. Combining liposomes without distinguishing specific types into one group with nanoparticles seems to be a mental shortcut too far. The authors essentially ignore the issues of toxicity and biodegradability which is not obvious in the case of metallic and oxide nanoparticles.
Authors: We appreciate the Reviewer for the critical evaluation of our manuscript. Accordingly, we have revised our manuscript addressing the comments from the reviewer and the potential readers to understand.
Although, we certainly agree that nanoparticle-based drug delivery review work is not a new topic, but nanoparticle-based drug delivery for vascular application is certainly a new topic and worth consideration for a review work. Hence, in this review work, we have tried to focus on nanoparticle-based drug delivery for vascular application without diverging into other direction such as anti-cancer drug delivery. Moreover, we have incorporated and discussed a section where nanoparticle’s behaviour in blood vessels which directly affects the outcome of the nanoparticle for its targeted delivery in vascular applications. Similarly, required factors of nanomaterials for drug delivery was discussed in terms of vascular applications only.
Regarding the nanoparticle division section, we wanted to make it a brief section as there is plethora of review articles which focuses on nanoparticle division for drug delivery in detail. However, the focus of our review manuscript is nanoparticle-based drug delivery for vascular application and hence we have always maintained our focus on that direction only. Nevertheless, we have also added the nanoparticle division in brief so that it was for potential readers to understand the overall concept of the review work. Moreover, we also agree with the reviewer that liposome nanoparticles are worthy of a different section for elaborate discussion, but it would have lengthened the manuscript and may have diverted the attention of the potential readers from the focus of the manuscript. Hence, we have given a brief description of all the nanoparticles for drug delivery specifically for vascular applications.
Regarding the toxicity and biodegradability of metal oxide nanoparticles, we have highlighted the portion (section 2 and section 6.4) in blue colour where we have discussed about the toxicity, biocompatibility and biodegradability of metal oxide nanoparticle. Please see the revised manuscript for this purpose. So, we sincerely hope the reviewer will agree with these points along with the changes and publish our manuscript in its revised version.

Reviewer 2 Report
Comments and Suggestions for Authors
Writing rviews is difficult because, operating on data that already exists, it is necessary to show something new. I don't see it in the authors' work. Even such basic things as the division of nanoparticles is chaotic and controversial.Combining liposomes without distinguishing specific types into one group with nanoparticles seems to be a mental shortcut too far. The authors essentially ignore the issues of toxicity and biodegradability which is not obvious in the case of metallic and oxide nanoparticles.
Author Response
Reviewer 2
General statement
Writing reviews is difficult because, operating on data that already exists, it is necessary to show something new. I don't see it in the authors' work. Even such basic things as the division of nanoparticles is chaotic and controversial. Combining liposomes without distinguishing specific types into one group with nanoparticles seems to be a mental shortcut too far. The authors essentially ignore the issues of toxicity and biodegradability which is not obvious in the case of metallic and oxide nanoparticles.
Authors: We appreciate the Reviewer for the critical evaluation of our manuscript. Accordingly, we have revised our manuscript addressing the comments from the reviewer and the potential readers to understand.
Although, we certainly agree that nanoparticle-based drug delivery review work is not a new topic, but nanoparticle-based drug delivery for vascular application is certainly a new topic and worth consideration for a review work. Hence, in this review work, we have tried to focus on nanoparticle-based drug delivery for vascular application without diverging into other direction such as anti-cancer drug delivery. Moreover, we have incorporated and discussed a section where nanoparticle’s behaviour in blood vessels which directly affects the outcome of the nanoparticle for its targeted delivery in vascular applications. Similarly, required factors of nanomaterials for drug delivery was discussed in terms of vascular applications only.
Regarding the nanoparticle division section, we wanted to make it a brief section as there is plethora of review articles which focuses on nanoparticle division for drug delivery in detail. However, the focus of our review manuscript is nanoparticle-based drug delivery for vascular application and hence we have always maintained our focus on that direction only. Nevertheless, we have also added the nanoparticle division in brief so that it was for potential readers to understand the overall concept of the review work. Moreover, we also agree with the reviewer that liposome nanoparticles are worthy of a different section for elaborate discussion, but it would have lengthened the manuscript and may have diverted the attention of the potential readers from the focus of the manuscript. Hence, we have given a brief description of all the nanoparticles for drug delivery specifically for vascular applications.
Regarding the toxicity and biodegradability of metal oxide nanoparticles, we have highlighted the portion (section 2 and section 6.4) in blue colour where we have discussed about the toxicity, biocompatibility and biodegradability of metal oxide nanoparticle. Please see the revised manuscript for this purpose. So, we sincerely hope the reviewer will agree with these points along with the changes and publish our manuscript in its revised version.
